# Quantum Bohmian-Inspired Potential to Model Non–Gaussian Time Series and Its Application in Financial Markets

**DOI:** 10.3390/e25071061

**Published:** 2023-07-14

**Authors:** Reza Hosseini, Samin Tajik, Zahra Koohi Lai, Tayeb Jamali, Emmanuel Haven, Reza Jafari

**Affiliations:** 1Department of Physics, Shahid Beheshti University, Evin, Tehran 1983969411, Iran; 2Physics Department, Brock University, St. Catharines, ON L2S 3A1, Canada; 3Department of Physics, Islamic Azad University, Firoozkooh Branch, Firoozkooh 3981838381, Iran; 4Porous Media Research Lab, Department of Geology, Kansas State University, Manhattan, KS 66506, USA; 5Faculty of Business Administration, Memorial University of Newfoundland, St. John’s, NL A1C 5S7, Canada; ehaven@mun.ca; 6Institute of Information Technology and Data Science, Irkutsk National Research Technical University, Lermontova St., 664074 Irkutsk, Russia; 7Center for Communications Technology, London Metropolitan University, London N7 8DB, UK

**Keywords:** non-Gaussian time series, MRW, Bohmian quantum

## Abstract

We have implemented quantum modeling mainly based on Bohmian mechanics to study time series that contain strong coupling between their events. Compared to time series with normal densities, such time series are associated with rare events. Hence, employing Gaussian statistics drastically underestimates the occurrence of their rare events. The central objective of this study was to investigate the effects of rare events in the probability densities of time series from the point of view of quantum measurements. For this purpose, we first model the non-Gaussian behavior of time series using the multifractal random walk (MRW) approach. Then, we examine the role of the key parameter of MRW, λ, which controls the degree of non-Gaussianity, in quantum potentials derived for time series. Our Bohmian quantum analysis shows that the derived potential takes some negative values in high frequencies (its mean values), then substantially increases, and the value drops again for rare events. Thus, rare events can generate a potential barrier in the high-frequency region of the quantum potential, and the effect of such a barrier becomes prominent when the system transverses it. Finally, as an example of applying the quantum potential beyond the microscopic world, we compute quantum potentials for the S&P financial market time series to verify the presence of rare events in the non-Gaussian densities and demonstrate deviation from the Gaussian case.

## 1. Introduction

Since ancient times, mathematical and geometrical models have been adopted to study the world around us, and probability theories have been employed to deal with uncertainties of various events. However, in recent decades, some statistical experimental data in social science, notably in economics and psychology and mostly within the area of human decision making, have been observed to infringe the laws of classical probability. It has been proposed that the mathematical framework of quantum theory can offer some solutions to challenges of this kind, as the apparatus of quantum probability theory differs significantly from the classical one. Nowadays, we witness how societies, people, and events interact with one another on a global scale. Events happening in one corner of the globe can have a significant impact thousands of miles away. The behavior of the social and economic systems has transcended the classical framework. In the new decade, with the employment of some fundamental rules and laws from quantum theory, such as the loss of determinism, quantum superposition, and entanglement, physicists aim to uncover and predict the behavior of various systems in the macro-world. They strongly believe that, in studying some of the financial and social systems, violating the laws of classical probability, a deeper uncertainty principle, relative to the uncertainty represented by classical probability theory, exists [1,2]. The number of applications of quantum theory to social and financial problems, ranges from cognitive science and psychology, to economy, and quantum computing for finance [3,4,5,6,7,8]. In finance, more specifically, we note the quantum modeling of risks and decision-making as it can be applied to the analysis of financial markets [9,10,11,12,13,14,15]. According to these studies, the classical price dynamics can no longer be applied to all modern financial markets to study the price trajectories of these markets, and one also needs to consider the significant importance of multiple behavioral factors. Moreover, the fact that traders in modern financial markets tend to behave stochastically given their free wills needs to be taken into account.

Furthermore, as one referee of this paper remarked, there is a “growing importance of environmental noise in the dynamics of complex physical systems”. As the referee further indicates, this “positive role of noise and fluctuations in complex systems” may also be connected to the work of “Giorgio Parisi’s Nobel Prize in Physics (2021) on the importance of fluctuations on physics and also social systems” (see [16,17,18]).

We also agree with the said referee that market dynamics are out of equilibrium and Langevin dynamics can be appropriate. As the referee mentions “by investigating these types of models, counterintuitive effects due to volatility could reveal a new indicator of market stability” (see [19,20,21,22]).

One of the most prominent deficiencies of applying classical mechanics to uncover the behaviors of financial markets reveals itself in the non-locality-like features of the modern financial markets. In order to study modern financial markets, we need to consider the price return of a period, consisting of several days “entangled“ to each other. The quantum mechanical approach uses extended ideas based on quantum entanglement to examine the correlation of different time series with a high number of extreme events to study and predict their evolution in time. In this paper, we employed the Bohmian quantum potential method in order to study an example of these entanglements in financial markets and analyze the impacts of extreme events on these time series.

## 2. Bohmian Mechanics and the Quantum Potential Inspired Method

For the benefit of the reader, we would like to sketch some salient elements on the historical development of Bohmian mechanics. One referee of this paper made a very interesting historical overview. We cite it here as follows: “Bohmian mechanics is one of the coherent stochastic theories of quantum mechanics. The first theory was put forward by Fényes [23,24], who was able to show the Schrödinger equation could be understood as a kind of diffusion equation for a Markov process. Then, Louis de Broglie [25] felt compelled to incorporate a stochastic process underlying quantum mechanics to make particles switch from one pilot wave to another. The most recent and widely known theory was the stochastic mechanics by Edward Nelson [26,27,28]. Bohmian mechanics, which is also called the de Broglie-Bohm theory, the pilot-wave model, and the causal interpretation of quantum mechanics, is a version of quantum theory discovered by Louis de Broglie in 1927 and rediscovered by David Bohm in 1952. A system of particles is described in part by its wave function, evolving according to Schrödinger’s equation, and completed by the specification of the actual positions of the particles, evolving according to the “guiding equation”, which expresses the velocities of the particles in terms of the wave function”.

The Schrodinger wave function which is at the heart of quantum mechanics can be used to explain the relationship between events in space–time, and accordingly, address the obstacle of non-locality of events. The notion that the wave function of a system, evolving according to the Schrodinger equation, is interpreted as an active information field, builds the foundation of Bohmian mechanics, in which information, at the level of human perception, functions according to postulates from information at the quantum level. This approach is fundamentally based on the active information analysis of Bohmian mechanics and its applications to cognitive sciences [29,30]. The Schrodinger formalism demonstrates how Bohmian mechanics complies with our understanding of financial markets as an example of a correlated system [31,32]. We represent our financial pilot-wave ψ(q), evolving according to Schrodinger’s equation in the following form:(1)$$\psi \left(q,t\right)=R\left(q,t\right)e^{iS(q,t)/\hbar },$$
where R(q,t)=|ψ(q,t)| is the amplitude, and S(q,t) is the phase of the defined wave function ψ(q,t). Substituting the pilot wave into the Schrodinger equation yields:(2)$$\frac{\partial R^{2}}{\partial t}+\frac{1}{m}\frac{\partial (R^{2}\frac{\partial S}{\partial q})}{\partial q}=0,$$
(3)$$\frac{\partial S}{\partial t}+\frac{1}{2m}{\left(\frac{\partial S}{\partial q}\right)}^{2}+\left(V-U\right)=0,$$
where the Bohmian quantum potential is:(4)$$U=\frac{\hbar^{2}}{2mR}\frac{\partial^{2}R}{\partial q^{2}}.$$

In the following section, we demonstrate how we want to analyze the dynamics of non-Gaussian functions with the help of the Bohmian quantum potential. We first give a review on the multifractal process and multifractal random walk to model the non-Gaussian probability density function of our desired data.

## 3. Multifractal Formalism

Multifractality has previously been applied to consider the scale invariance features of various objectives in several areas of research. Different studies have examined the concept of pairing multifractality between time series based on the increasing number of rare events that can generate deviation from Gaussian density [33,34,35]. In particular, the MRW models, popular in modeling stock fluctuations in the financial market, have become the focus of some recent analyses [36,37]. Moreover, the relationship between turbulence and finance has triggered the demand for employing multifractal models which were previously studied in the framework of multiplicative random cascades [38,39]. Here, we aim to employ multifractal concepts to account for the scale-invariant properties of financial data, based on which the robust technique of MRW is introduced and applied [40].

The non-Gaussian probability density function with the robust multifractality arises from the strong log-normal deviation from the normal state which is primarily due to the occurrence of large fluctuations in the data set. The exact multifractal properties are a consequence of the correlation in the argument of the logarithm of the stochastic variances [41].

Consider a stochastic process, represented by X(t), which may be a function of space–time, the increment fluctuations of the data sets at a time-scale τ can be shown as:(5)$$\Delta X_{\tau }\left(t\right)=X\left(t+\tau \right)-X\left(t\right).$$
The process is called scale-invariant when the absolute moment M(q) has the following power-law behavior:(6)$$M\left(q,\tau \right)=\langle {\left|\Delta X\left(\tau \right)\right|}^{q}\rangle =M\left(q,\tau \right)\propto \tau^{\xi_{q}},$$
where we define $\xi_{q}$ as the exponent of the power law, which is responsible for characterizing the scale invariance properties of the fractal function [42] and can be shown by:(7)$$\xi_{q}=qH-1/2\left(q\left(q-2\right)\lambda^{2}\right).$$

The process is then called monofractal if $\xi_{q}$ is a linear function of *q*, and multifractal if $\xi_{q}$ is a nonlinear function of *q* [43]. In Equation (Equation 7), *H* is the Hurst scaling exponent of the time series, such that 0<H<1. For 0<H<0.5, the system is known to be anti-correlated, 0.5<H<1 leads to correlation, and for H=0, we have an uncorrelated system. The value of λ scales the non-Gaussianity, such that for the Gaussian case, we have: λ=0, corresponding to $\xi_{q}\propto q$, and indicating the fractality of the signal. For the non-Gaussian case, with the λ≠0 signal will represent multifractal behavior. In this section, using only a set of few variables, we apply multifractal statistics to model the increment in fluctuations of the data [44,45].

For a log-normal cascade at the smallest scale Δt and for each time-lag, τ, to obtain a good candidate that satisfies the cascading relation, known as the MRW, we write:(8)$$\Delta_{\tau }X\left(t\right)=\epsilon \left(t\right)e^{\omega (t)},$$
where ϵ(t), and ω(t) are Gaussian variables with their corresponding variances being indicated by $\sigma^{2}$ and $\lambda^{2}$, respectively. In this approach, regarding the stated stochastic variables, in order to convey the analytical calculation of the quantum potential for the stated model, we first need to define the non-Gaussian probability density function (PDF) with fat-tail time series. Hence, we can find the relationship between non-Gaussian parameters, λ and multifractality, which comes from the nonlinear function’s general exponents $\xi_{q}$ vs. *q*. Based on the Castaing model, a process is called self-similar if the increment’s probability density functions at scales τ are related by the following equation [12,46]
(9)$$P\left(\Delta_{\tau }X\right)=\int G_{\tau }\left(Ln\sigma \right)\frac{1}{\sigma }F_{\tau }\left(\frac{\Delta_{l}x}{\sigma }\right)dln\sigma ,$$
where
(10)$$G\left(Ln\sigma \right)=\frac{1}{\sqrt{2\pi }\lambda }exp\left(-\frac{ln^{2}\sigma }{2\lambda^{2}}\right)$$
(11)$$F\left(\frac{\Delta_{\tau }x}{\sigma }\right)=\frac{1}{\sqrt{2\pi }}exp\left(-\frac{{\left(\Delta_{\tau }x\right)}^{2}}{2\sigma^{2}}\right).$$

The Bohmian quantum potential depends only on the second spatial derivative of the amplitude of the wave, taking $P=R^{2}$, we write:(12)$$\frac{d^{2}P\left(x\right)}{dx^{2}}=\frac{1}{\pi \lambda }\int_{0}^{\infty }\frac{\left(2x^{2}-\sigma^{2}\right)}{\sigma^{5}}e^{-\frac{\ln^{2}\frac{\sigma }{\sigma_{0}}}{2\lambda^{2}}}e^{-\frac{x^{2}}{\sigma^{2}}}\left(dln\sigma \right).$$

We now compute the quantum potential ($U\propto \frac{1}{P(x)}\frac{d^{2}P\left(x\right)}{dx^{2}}$) for the above probability density function (without considering power two of R) and plot them for a range of parameters for further comparison.

## 4. Results for Computing Quantum Potential and Real Data Fit

Based on our previous arguments, employing Bohmian mechanics and studying the strong financial effects on the market trajectories, we aim to describe the dynamics of the financial pilot wave. With the use of probability density function defined as Equation (Equation 9) for non-Gaussian functions with a range of λ’s, we examine the effects of extreme events, and compute their corresponding quantum potentials to analyze and fit real market data subsequently. In our calculation, we consider *ℏ* as a price scaling parameter and will assume ℏ=1. In the up panel of Figure 1, we compare the contribution of rare events on the quantum potential for Gaussian density (λ=0) and non-Gaussian density functions with several $\lambda^{\prime }s$. The up panel of Figure 1 shows some density functions for σ=1 and a range of λs, and the bottom panel displays the corresponding calculated quantum potentials.

As can be noted from the graph, increasing the λ leads to a remarkable distortion promoting the occurrence of the extreme events. Moreover, this figure shows how this increase in the value of λ will induce a potential barrier, which, when transcended, will provoke the system to proceed to its critical state. Our quantum potential analysis shows that there exist two main contributions to this potential. Firstly, one can discuss the vicinity of the mid-value where the system holds a low quantum potential, revealing the system’s tendency to survive in this region. Upon increasing the non-Gaussianity parameter, we remarked that the quantum potential which would increase and act as a potential well is now shrinking. This reveals its contribution to the occurrence of the influential events in the system. However, with the further increase in the non-Gaussian parameter, we observed that the quantum potential for these big events compared to the mean is reduced. The presence of these two phases uncovers the aim of the system to stay in the region where extreme events take place.

As was expected, the behavior of a quantum barrier, revealing the limitation that the price range faces, explains how increasing the non-Gaussian parameter λ, in spite of the rise in the depth of the well, does not induce infinity in the well. From this behavior, we concluded that any market whose energy is large enough to surpass this barrier will enter its critical phase. Next, following the multifractal random walk formalism, we constructed a toy model with λ=0.3 and σ=1, as shown in Figure 2, and calculate the corresponding quantum potential for our model. As can be remarked from the graph, for this λ, the energy of the model exceeds the potential barrier, and consequently, the system undergoes its critical condition. To analyze the real market and compare it to our toy model, we also illustrated the behavior of the Nikkei index from 1990 to 2020 in the same figure and examined it. The market PDF, with σ=0.802, shows how it sustains its critical stage with the average λ of 0.399. As we noted earlier, the analysis of the quantum potential of the Nikkei index unveils that, compared to Gaussian density functions, the density functions of these markets contain plenty of sudden fluctuations. The main difference between our model and the real data manifests itself in the value for the return. As denoted from Figure 2, unlike the behavior of our constructed toy model, the real market return function tends to behave in-homogeneously. This deviation comes from the fact that, in our model, λ is set to a uniform value. Nonetheless, real markets with the same average value of λ sustain a relatively non-stationary character. Considering the inconsistencies, λ in these series would have some time dependency, and therefore, we require the consideration of an average value for our analysis.

## 5. Conclusions

Various research studies have been utilizing models and techniques to apply the laws of a quantum system to the macro world’s social systems and financial markets. Moreover, one of the most distinctive features of financial market data lies in their non-Gaussian performance and the occurrence of the probability of extreme events. Upon the question of whether this behavior stems from an underlying structural pattern, we employed an innovative method to analyze these functions in the framework of quantum theory. Thus, by implementing quantum Bohmian mechanics and deriving the corresponding quantum potentials for the non-Gaussian density function of the data, we can identify the distinction between the behavior of extreme events in the financial markets explained by Gaussian and non-Gaussian functions. Additionally, we confirmed how these extreme events in the non-Gaussian cases introduce themselves via a potential barrier in a process, aiming to linger in its mean values. However, we cannot miss the fact that the occurrence of extreme events has a significant role in these processes. Consequently, in this work, we verified the application of using the quantum potential for non-Gaussian functions to clarify the behavior of financial markets. Nevertheless, we firmly believe that there is a wide range of underlying applications to this approach for a variety of natural processes. In further work, we aim to consider the Bohmian mechanics approach relative to financial price path generation. It would be very interesting to see how the behavior of the real and quantum potential functions can influence such a trajectory. The more we can understand the behavior of the potentials relative to finance, the better we can understand how such objects can influence price dynamics. The promise for new financial modeling could be enormous.

## Figures and Tables

**Figure 1 entropy-25-01061-f001:**
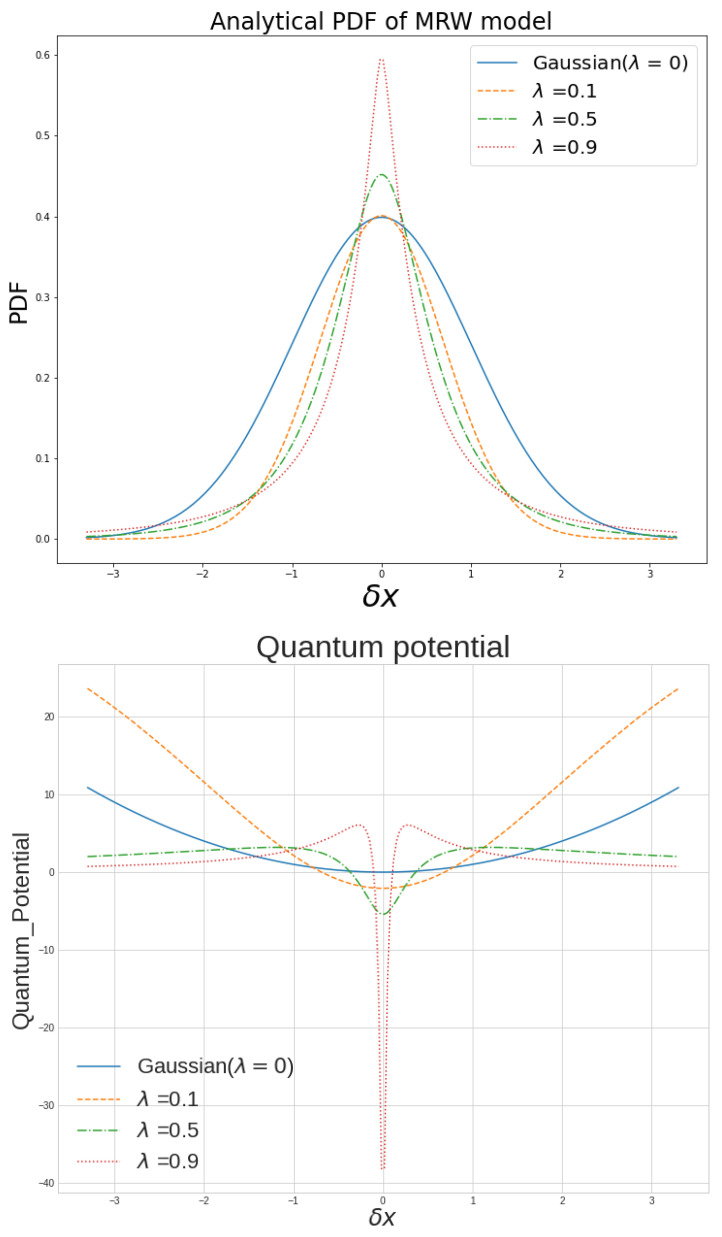
(**Panel up**) The density function for the cascade model with three different non-Gaussian parameter values λ∈[0.1,0.5,0.9] and σ=1. We also compare it with Gaussian density function with σ=1. (**Panel bottom**) The figure shows the corresponding derived quantum potential.

**Figure 2 entropy-25-01061-f002:**
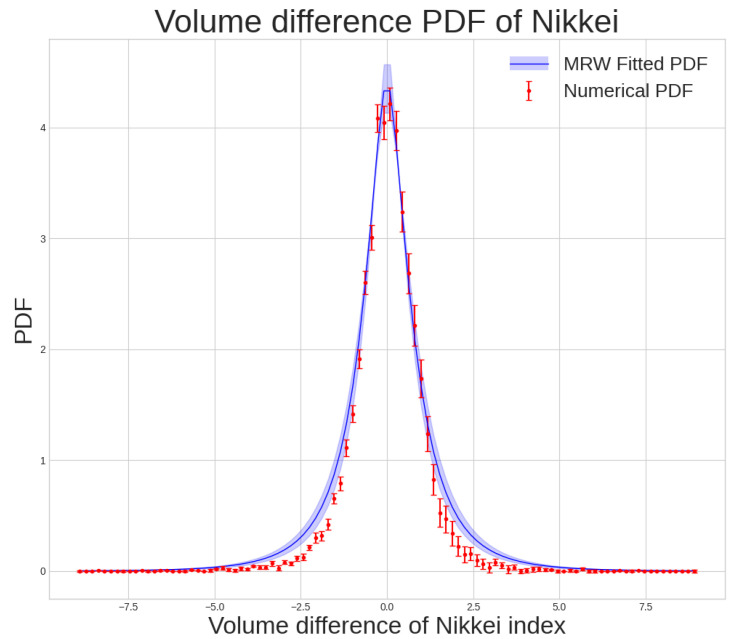
The left panel shows the PDF of the return price of Nikkei index data from 1990 to 2021 and the PDF of the MRW fit, and the right panel presents the corresponding quantum potential for Nikkei.

## Data Availability

Data sets are available in finance.yahoo.com/.

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
