# Peer review of "Quantum Bohmian-Inspired Potential to Model Non–Gaussian Time Series and Its Application in Financial Markets"

_entropy, 2023, doi:10.3390/e25071061_

Round 1
Reviewer 1 Report
See the attached report.

Reviewer 2 Report
I would like to recommend the article for publication in this special issue. Based on the increasingly more prominent view of a greater effectiveness of quantum-like modeling (based on the formalism of quantum theory) vs. those of classical-like modeling, the article offers as promising application of Bohmian formalism (which is different from that of the standard quantum mechanics [QM]), to the financial markets. This application, it might observed, is not affected by difficulties and controversies accompanying Bohmian theory in physics, because these difficulties are physical in nature, and are not factors in using the mathematical formalism of Bohmian theory beyond physics. In the field of quantum-like modeling, it is unusual in that most of the work in the area uses the formalism of the standard QM. The actual findings of the article are promising and convincing in showing the value of non-Gaussian functions in analysis the data in question. Section 4 and the Conclusion could be extended a bit to reflect on a potential of Bohmian quantum-like models in other areas of economics and finance. There are several minor typos. Thus change “The number of applications from applying quantum theory to social and financial problems, ranges from cognitive science and psychology, to economy, and quantum computing for finance” (p. 2) to “The number of applications of quantum theory to social and financial problems ranges from cognitive science and psychology, to economy, and quantum computing for finance.”
English is fine!
Round 2
Reviewer 1 Report
The following references should be corrected:
[18] B Lisowski, B.; D Valenti, D.; B Spagnolo, B.; M Bier, M.; E Gudowska-Nowak, E. Stepping molecular motor amid Lévy white noise. Phys. Rev. E 2015, 91, 042713.
[20] Bonanno, G.; Spagnolo, B. Escape times in stock markets, Fluctuation and Noise Letters 2005, 5, L325-L330.
Author Response
We have corrected references [18] and [20] based on their journal reports.
But we can not upload the Tex file. So we have added the correct references as follows.
[18] Stepping molecular motor amid Lévy white noise
Bartosz Lisowski, Davide Valenti, Bernardo Spagnolo, Martin Bier, and Ewa Gudowska-Nowak
Phys. Rev. E 91, 042713 – Published 24 April 2015
https://journals.aps.org/pre/abstract/10.1103/PhysRevE.91.042713
[20] Noise in Condensed Matter and Complex Systems;
G. BONANNO and B. SPAGNOLO
Fluctuation and Noise Letters, Vol. 05, No. 02, pp. L325-L330 (2005)
https://doi.org/10.1142/S0219477505002720
https://www.worldscientific.com/doi/10.1142/S0219477505002720